# Platelet Mitochondrial Respiration, Endogenous Coenzyme Q_10_ and Oxidative Stress in Patients with Chronic Kidney Disease

**DOI:** 10.3390/diagnostics10030176

**Published:** 2020-03-23

**Authors:** Anna Gvozdjáková, Zuzana Sumbalová, Jarmila Kucharská, Mária Komlósi, Zuzana Rausová, Oľga Vančová, Monika Számošová, Viliam Mojto

**Affiliations:** 1Pharmacobiochemical Laboratory of 3rd Department of Internal Medicine, Faculty of Medicine, Comenius University in Bratislava, Sasinkova 4, 811 08 Bratislava, Slovakia; zuzana.sumbalova@fmed.uniba.sk (Z.S.); jarmila.kucharska@fmed.uniba.sk (J.K.); zuzana.rausova@fmed.uniba.sk (Z.R.); olga.vancova@fmed.uniba.sk (O.V.); 2Faculty of Medicine, 3rd Department of Internal Medicine, Comenius University in Bratislava, 4, 811 08 Bratislava, Slovakia; komlosi.mar@gmail.com (M.K.); monika.szamosova@gmail.com (M.S.); viliam.mojto@gmail.com (V.M.)

**Keywords:** platelets, mitochondria, respiration, chronic kidney disease, coenzyme Q_10_, oxidative stress

## Abstract

Chronic kidney disease (CKD) is characterized by a progressive loss of renal function and a decrease of glomerular filtration rate. Reduced mitochondrial function, coenzyme Q_10_ (CoQ_10_), and increased oxidative stress in patients with CKD contribute to the disease progression. We tested whether CoQ_10_ levels, oxidative stress and platelet mitochondrial bioenergetic function differ between groups of CKD patients. Methods: Twenty-seven CKD patients were enrolled in this trial, 17 patients had arterial hypertension (AH) and 10 patients had arterial hypertension and diabetes mellitus (AH and DM). The control group consisted of 12 volunteers. A high-resolution respirometry (HRR) method was used for the analysis of mitochondrial bioenergetics in platelets, and an HPLC method with UV detection was used for CoQ_10_ determination in platelets, blood, and plasma. Oxidative stress was determined as thiobarbituric acid reactive substances (TBARS). Results: Platelets mitochondrial respiration showed slight, not significant differences between the groups of CKD patients and control subjects. The oxygen consumption by intact platelets positively correlated with the concentration of CoQ_10_ in the platelets of CKD patients. Conclusion: A decreased concentration of CoQ_10_ and oxidative stress could contribute to the progression of renal dysfunction in CKD patients. The parameters of platelet respiration assessed by high-resolution respirometry can be used only as a weak biological marker for mitochondrial diagnosis and therapy monitoring in CKD patients.

## 1. Introduction

The mortality rate of patients with chronic kidney diseases (CKDs) increases with a decrease in the glomerular filtration rate (eGFR). The development and progression of CKD are closely linked with non-communicable diseases (NCDs) like cardiovascular diseases, cancers, respiratory tract diseases, and diabetes [1]. The most common causes of CKD include diabetes, hypertension, atherosclerosis, glomerulonephritis, and polycystic kidneys [2,3]. Patients with CKD have a high risk of death from stroke or heart attack [4].

CKD is characterized by a progressive loss of renal function, in the presence or absence of albuminuria, that is caused by chronic inflammation, oxidative stress and vascular remodeling [4]. Other factors such as diabetes, changes in cellular energy metabolism, mitochondrial dysfunction, and reduced antioxidants protection including coenzyme Q_10_ (CoQ_10_) concentration contribute to CKD [5,6]. In diabetic kidney disease (DKD), increased plasma glucose levels may directly induce mitochondrial injury in renal tubular cells, podocytes, mesangial cells, and glomerular endothelial cells. Mitochondrial dysfunction and the overproduction of free oxygen radicals play a pivotal role in the development and progression of DKD [7]. Coenzyme Q_10_ is an essential component of the mitochondrial respiratory chain for energy production in the form of adenosine triphosphate (ATP) via oxidative phosphorylation [8]. As an antioxidant, CoQ_10_ quenches free radicals and directly prevents lipid peroxidation [9]. Different studies have demonstrated that platelet mitochondrial dysfunction has also been used for studying mitochondria-related diseases [10,11,12,13,14,15,16,17,18].

We tested the hypothesis of whether there is a correlation between CoQ_10_ levels and platelet mitochondrial function in CKD patients, as well as if the high-resolution respirometry (HRR) method is suitable for the detection of small differences in mitochondrial bioenergetic function in circulating platelets in subgroups of CKD patients with arterial hypertension (CKD+AH) or arterial hypertension and diabetes mellitus (CKD+AH+DM).

## 2. Materials and Methods

### 2.1. Subjects

Twenty-seven patients with CKD participated in the study: 19 men and 8 women, aged from 36 to 72 years with a mean age of 58.3 ± 1.68 years and a mean body mass index (BMI) of 27.4 ± 1.11 kg m^−2^. All chronic CKD patients were treated with relevant conventional therapy. From these CKD patients, 17 patients with AH (CKD+AH) were treated with conventional therapy for cardiovascular diseases—their mean age was 54.9 ± 2.20 years and their mean BMI was 26.9 ± 1.39 kg m^−2^—and 10 patients with DM (CKD+AH+DM) were treated with conventional therapy for arterial hypertension and for diabetes Type 2—their mean age was 64.0 ± 1.13 years and their mean BMI was 28.3 ± 1.79 kg m^−2^. The control group consisted of 12 human subjects (3 men and 9 women) without medicals therapy, aged from 56 to 81 years, with a mean age of 67.8 ± 2.05 years and a mean BMI of 23.9 ± 0.6 kg m^−2^. The physical characteristics of human volunteers and all CKD patients are summarized in Appendix A.

The study was carried out according to the principles expressed in the Declaration of Helsinki, and the study protocol was approved by the Ethical Committee of the Academic Ladislav Dérer’s Hospital, Bratislava, Slovakia (1/0245/19, 27 June 2018). Written informed consent form was obtained from each subject before enrollment in the study.

### 2.2. Observed Parameters

The following parameters were measured in clinical biochemical laboratory using standard methods: Kidney parameters included serum creatinine, uric acid, and estimated glomerular filtration rate (eGFR). Furthermore, blood hemoglobin, glucose, and serum c-reactive proteins were determined. Lipids parameters included: triacylglycerols (TAG), low density lipoprotein (LDL)-cholesterol, high density lipoprotein (HDL)-cholesterol, and total cholesterol. Liver function tests included the activities of liver enzymes like AST (aspartate aminotransferase), ALT (alanine aminotransferase), and GMT (gama-glutamyltransferase). The metabolic characteristics of the human volunteers and the groups of CKD patients are summarized in Table 1 and in Appendix A.

### 2.3. Platelets Preparation

Blood samples were collected by venipuncture. For platelet (PLT) isolation, 18 mL of venous blood was added to K_3_EDTA (tripotassium ethylenediaminetetraacetic acid) tubes each day between 7:00 and 8:00 a.m. Blood samples were transported at 25 °C room temperature to the laboratory and centrifuged at room temperature at 200× *g* for 10 min using swing-out rotor without breaking. Platelet-rich plasma (PRP) was transferred into a new plastic tube and mixed with 100 mM EGTA (ethylene glycol-bis(2-aminoethylether)-N,N,N′,N′-tetra-acetic acid) to a final concentration of 10 mmol·L^−1^. The pellet, after centrifugation at 1200× *g*, was washed with 4 mL of DPBS (Dulbecco’s Phosphate-Buffered Saline) plus 10 mM EGTA and finally resuspended in 0.4 mL of the same solution. The PLT suspension was counted (10 times diluted) on hematological analyzer Mindray BC-2800 (Mindray, China) [19].

### 2.4. Platelets Mitochondrial Respiration and Oxidative Phosphorylation

Oxygen consumption in the intact and permeabilized platelets and the capacity of oxidative phosphorylation (OXPHOS) at Complex I were determined with the HRR method. HRR is a sensitive technique to determine mitochondrial bioenergetic function in human platelets that are isolated from peripheral blood [20,21]. For mitochondrial respirometric analysis, 200 × 10^6^ platelets were used in a 2 mL chamber of an O2k-Respirometer (Oroboros Instruments, Innsbruck, Austria). The respiration was measured at 37 °C in a mitochondrial respiration medium, MiR05 [20] and 20 mM creatine under continuous stirring at 750 rpm. The data were collected with the DatLab software (Oroboros Instruments, Innsbruck, Austria) with a data recording interval of 2 s.

### 2.5. Experimental Protocol for Platelet Mitochondrial Respiration

For the evaluation of mitochondrial function in isolated platelets, the modified SUIT (substrate-uncoupler-inhibitor-titration) reference protocol (RP) 1 [22] was used (Figure 1). The protocol started with the addition of 200 × 10^6^ platelets into the O2k chamber with a respiration medium equilibrated at 37 °C. After stabilization at routine respiration, digitonin was added at final concentration 20–30 μg·mL^−1^ for the permeabilization of plasma membrane, which was accompanied by a decline in the respiration rate close to zero. Then, Complex I (CI)-linked substrates pyruvate (5 mM) and malate (2 mM) were added for the evaluation of CI-linked LEAK [20] (analogous to State 4) respiration. In the next step, adenosine diphosphate (ADP) at a saturating concentration (1 mM) was added for evaluation of OXPHOS capacity driven by CI-linked substrates. After that, cytochrome *c* (5 mM) was added to test the integrity of the outer mitochondrial membrane, and subsequently an uncoupler carbonyl cyanide m-chlorophenyl hydrazine (CCCP) was titrated to the chamber (0.5 μM steps) until maximum respiration corresponding to maximal capacity of electron transfer (ET) with the given CI-linked substrates was reached. An reduced nicotinamide adenine dinucleotide (NADH)-linked pathway was additionally stimulated by CI-linked substrate glutamate (10 mM). The subsequent addition of the Complex II (CII)-linked substrate succinate (10 mM) supported the CII-linked electron transfer into the Q-junction for the determination of ET-capacity with convergent input through both CI and CII. In the next step, CI was inhibited by rotenone (1 μM) for the evaluation of CII-linked ET-capacity. Finally, the addition of Complex III (CIII) inhibitor antimycin A (2.5 μM) blocked the transfer of electrons through the respiratory chain and enabled the evaluation of the residual oxygen consumption that was not associated with electron transfer pathways; this was subtracted from all respiratory rates for the determination of mitochondrial electron transfer pathways-related oxygen consumption. The representative trace from the respirometric measurement is shown in Figure 1.

Step: Meaning.

ce1: Oxygen consumption rate of intact PLT (routine respiration).

1Dig: Respiration rate of mitochondria in PLT permeabilized with digitonin.

1PM: Complex I-linked LEAK (State 4) respiration reflects the rate of mitochondrial respiration with exogenous substrates (pyruvate and malate).

2D: Complex I-linked OXPHOS (State 3) after ADP addition reflects CI-linked ATP production.

2c: Cytochrome c addition—a test for the integrity of outer mitochondrial membrane.

3U: The rate after addition of an uncoupler CCCP represents maximal CI-linked oxidative capacity with substrates pyruvate and malate (uncoupled from OXPHOS).

4G: Noncoupled Complex I-linked oxygen consumption after the addition of substrate glutamate.

5S: Noncoupled Complex I- and Complex II-linked oxygen consumption after the addition of CII substrate succinate.

6Rot: Noncoupled Complex II-linked oxygen consumption after the addition of rotenone—an inhibitor of Complex I.

7Ama: Residual oxygen consumption (ROX) after the addition of antimycin A—an inhibitor of CIII represents respiration that is not associated with electron transfer pathways. This respiration is subtracted from all values for the determination of mitochondrial electron transfer pathways-related oxygen consumption.

### 2.6. Coenzyme Q_10-TOTAL_

Coenzyme Q_10-TOTAL_ (ubiquinol and ubiquinone) in isolated platelets, whole blood, and plasma was determined with the HPLC method with UV detection [23,24], modified by the authors [25]. CoQ_10-TOTAL_ concentrations were determined after the oxidation of ubiquinol with 1,4-benzoquinone [26]. Isolated platelets (100–200 μL) were disintegrated with 500 μL of cold methanol. The cell suspension was extracted with 2 mL of hexane by shaking for 5 min. After centrifugation at one thousand g for 5 min, the organic layer was separated and evaporated under nitrogen. Blood and plasma samples (500 μL) were extracted with a hexane/ethanol mixture (5/2 *v/v*) two times. The residues were taken up in ethanol and injected into a reverse phase HPLC column. Elution was performed with methanol/acetonitrile/ethanol (6/2/2 *v/v/v*). The concentrations of CoQ_10_ were detected with a UV-detector at 275 nm, using an external standard. Data were collected and processed with a CSW32 chromatographic station. Concentrations of CoQ_10-TOTAL_ in platelets were calculated in pmol · 10^−9^ cells, and they were calculated in blood and plasma in μmol · L^−1^.

One oxidative stress marker—thiobarbituric acid reactive substances (TBARS) in plasma—was estimated with a spectrophotometric method [27].

### 2.7. Data Analysis

All data were analyzed with an unpaired Student’s *t*-test for evaluation differences in parameters between the groups of CKD patients vs. control subjects. The level of statistical significance was set at *p* < 0.05. The results are expressed as mean ± SEM, and in % vs. values in the control group. Pearson’s correlation analyses were performed in GraphPad Prism 6.

## 3. Results

### 3.1. Metabolic Parameters of Healthy Volunteers and Patients with Chronic Kidney Disease

The metabolic parameters of healthy volunteers and patients with CKD are shown in Table 1. Regarding kidney parameters, eGFR was significantly lower (*p* < 0.001) in the groups of CKD-ALL; CKD+AH; and CKD+AH+DM type 2 patients in comparison with control data. Creatinine concentration in serum was significantly higher in the groups of CKD-ALL; CKD+AH (*p* < 0.001); and CKD+AH+DM (*p* < 0.01) in comparison with control data. Uric acid concentration was significantly higher in the groups of CKD-ALL and CKD+AH+DM (*p* < 0.05); in the group of CKD+AH, the increase was not significant (Table 1a). The concentration of hemoglobin was slightly (n. s.) decreased in all groups of CKD patients, and the concentration of the c-reactive protein (CRP) in all CKD groups (CKD-ALL; CKD+AH; and CKD+AH+DM) was significantly higher vs. the control group. Glucose concentration was slightly (n. s.) increased in the CKD-ALL and CKD+AH+DM patients (Table 1b). There were subtle (n. s.) changes in the lipids parameters in all groups of CKD patients vs. the control values: an increase in TAG concentration and the decrease in LDL- and HDL-cholesterol concentration. The concentration of the total cholesterol was slightly (n. s.) increased in the CKD+AH+DM group (Table 1c). The liver enzymatic activities of AST were slightly increased and the activities of GMT were slightly (n. s.) decreased in all groups of CKD patients vs. the control group. The activity of ALP was significantly increased in the CKD-ALL group (*p* < 0.001); the CKD+AH group (*p* < 0.05); and the CKD+AH+DM group (*p* < 0.05) in comparison with the control group (Table 1d); see Appendix A.

### 3.2. Platelet Mitochondrial Respiration and Oxidative Phosphorylation in Control Subjects and Groups of CKD Patients

In our trial, we did not find significant differences in platelet mitochondrial bioenergetic parameters between the control subjects and the groups of CKD patients (Figure 2 and Appendix A).

We present the slight differences in the parameters of platelet mitochondrial respiration in CKD groups in % vs. control group. The oxygen consumption of intact platelets (step ce1) in all CKD groups was similar to that of the control group. The rate of mitochondrial respiration with CI-linked substrates (State 4, step 1PM) in the group of CKD-ALL decreased to 86.2%; in the group of CKD and AH, it decreased to 80.3%; and in the group of CKD+AH+DM, it increased to 108.1% of the control values. CI-linked respiration coupled with ATP production (CI-linked OXPHOS, step 2D) was only slightly increased in the groups of CKD-ALL and CKD+AH. In the group of CKD+AH+DM, this parameter was increased to 123.6% of the control data. The test for the integrity of the outer mitochondrial membrane (the addition of cytochrome *c*) did not show difference between groups. The respiration after this step (step 2c) was increased in the group of CKD-ALL to 112.2%; in the group of CKD+AH, it increased to 111.3%; in the group of CKD+AH+DM, it increased to 127.1% of the control data. Maximal oxidative capacity after uncoupler titration (step 3U) in the group of CKD-ALL was 110.7% of the control values; in the group of CKD+AH, it was 109.3% of the control values; and in the group of CKD+AH+DM, it was 124.7% of the control values. We received similar results after addition of exogenous substrate glutamate (step 4G). This parameter in the CKD-ALL group constituted 106.3% of the control values; in the CKD+AH group, it was 105.5% of the control values; and in the CKD+AH+DM group, it was 119.4% of the control values. Noncoupled CI and II respiration (step 5S) in the CKD-ALL and CKD+AH groups was similar to that in the control group; in the CKD+AH+DM group, it was slightly higher (107.7% of the control values). The respiration after the inhibition of CI with rotenone (CII-linked noncoupled respiration, step 6Rot) was increased to 107.4% of the control values in the CKD+AH+DM group, and this parameter was similar to the control values in the CKD-ALL and CKD+AH groups (Appendix A).

### 3.3. Endogenous Coenzyme Q_10-TOTAL_ in Platelets, Blood and Plasma in Control Subjects and Groups of CKD Patients

The concentration of CoQ_10-TOTAL_ (ubiquinone and ubiquinol) in the platelets of the CKD-ALL patients was significantly decreased (*p* < 0.009) to 74.2% of the control values; in the CKD+AH group, it was decreased to 64.5% of the control values (*p* < 0.00008); and in the CKD+AH+DM group, it was decreased to 85.5% (n. s.) of the control values. The concentration of *CoQ_10-TOTAL_* in the whole blood of the CKD-ALL group was slightly, not significantly decreased to 81.6% of the control data; in the CKD+AH group, it was decreased to 76.0% of the control data; and in the CKD+AH+DM, it was decreased to 87.2% of the control data. The concentration of *CoQ_10-TOTAL_* in the plasma of the CKD-ALL patients was significantly decreased to 69.8% of the control values (*p* < 0.01); in the CKD+AH group, it was decreased to 67.2% (*p* < 0.016); and in the CKD+AH+DM, it was decreased to 71.5% of the control values (*p* = ns) (Figure 3, Appendix A).

Next, we evaluated correlations between CoQ_10-TOTAL_ and the parameters of platelet mitochondrial function.

### 3.4. Correlations between CoQ_10-TOTAL_ in Platelets and Respiration of Intact Platelets in Control Subjects and CKD Patients

Oxygen consumption by intact platelets (step ce1) in CKD-ALL patients positively correlated with the CoQ_10-TOTAL_ concentration in platelets (*p <* 0.05), (Figure 4). The association between these two parameters had the same trend in the control; CKD+AH; and CKD+AH+DM groups, but it did not reach statistical significance.

### 3.5. Plasma Lipid Peroxidation of CKD Patients

TBARS concentration was increased in the CKD-ALL group to 108.0%; (5.09 ± 0.12 µmol L^−1^), not significantly vs. control data (4.71 ± 0.13 µmol L^−1^). In the CKD+AH group, it was increased to 5.10 ± 0.14 µmol L^−1^ (108.3%). In the CKD+AH+DM group, it was increased to 5.12 ± 0.23 µmol L^−1^ (108.7%), indicating an increased oxidative stress in all groups of CKD patients. We calculated correlations between TBARS concentration and CoQ_10-TOTAL_ in the plasma of CKD-ALL patients and the control groups.

### 3.6. Correlation Between TBARS and CoQ_10-TOTAL_ in Plasma in Control Subjects and CKD-ALL Patients

We did not find significant correlations between TBARS and CoQ_10-TOTAL_ in plasma in the CKD-ALL or control groups (Figure 5); nevertheless, in patients with CKD (CKD-ALL), there was a trend for negative correlation (*p* = 0.078) between these two parameters.

## 4. Discussion

The prevalence, development, and progression of CKD has reached epidemic proportions. CKDs are closely linked with non-communicable diseases, and they are responsible for 71% of all death in the world. Annually, 17.9 million people become ill with cardiovascular diseases, 9 million people become ill with cancers, 3.9 million people become ill with respiratory tract diseases, and 1.6 million people become ill with diabetes [1]. NCDs are non-infections and non-transmissible among people; they originate from a combination of genetic, physiological, and lifestyle factors.

Chronic inflammation, oxidative stress, and changes in cellular energy metabolism participate in reduced kidney function [11,12,13]. Mitochondrial dysfunction and reduced CoQ_10_ concentration contribute to dysfunctional platelet activity in diseases, such as neurological diseases, cardiovascular diseases, diabetes mellitus, sepsis, Alzheimer’s disease, Parkinson’s disease, and chronic kidney disease [5,14,15,16,17]. CoQ_10_ plays a key role in mitochondrial energy production, and it transfers electrons from Complex I and Complex II to Complex III along the respiratory chain in the inner mitochondrial membrane. The electrons pass via cytochrome *c* to Complex IV of the mitochondrial respiratory chain. During the electron transfer, a proton gradient is developed through the inner mitochondrial membrane (this is important for the ATP-synthase), and ATP is produced via oxidative phosphorylation [5,6]. The measurement of mitochondrial function for the diagnostics of kidney dysfunction is difficult. The diagnostic value of the measurement of respiratory chain enzyme activities in frozen tissue from renal biopsy has not been confirmed [17].

Platelets are attractive sources of mitochondria that can be obtained less invasively, compared to tissue biopsy. Platelets are small (2–4 µm), anucleate circulating cells, and their lifespan of 7–10 days is largely determined by the mitochondria. Healthy platelets contain between five and eight mitochondria in a cell. Platelets in resting time mainly receive energy from glycolysis, which produces about 60% of cellular ATP, and oxidative phosphorylation provides about 30%–40% of ATP [14]. Platelets are used for the assessment of organ-specific mitochondrial dysfunction that is relevant to clinical outcomes [28]. Platelet mitochondrial dysfunction has been demonstrated in different studies on mitochondria-related diseases [10,18,29,30,31]. The high-resolution respirometry method allows for the dynamic monitoring of mitochondrial function in human platelets.

In our trial, we used circulating platelets for the detection of changes of mitochondrial function in patients with CKD. We did not find significant changes in the respiration of intact platelets or the permeabilized platelets in these groups of CKD patients vs. control data (Figure 2, Appendix A). We expressed the slight differences in platelets mitochondrial parameters between CKD groups in %, when the data of control group were used as 100%.

A reduced concentration of CoQ_10-TOTAL_ in the platelets, whole blood, and plasma of CKD-ALL patients was demonstrated. Insufficient CoQ_10_ concentration participates in decreased mitochondrial ATP production [19]. Our results are in an agreement with a previous study that found an inverse relationship between CoQ_10_ and renal function in patients with chronic kidney diseases [32].

Slight differences in platelet mitochondrial function were found between the group of CKD patients with arterial hypertension and the group of CKD patients with arterial hypertension and diabetes type 2. CoQ_10-TOTAL_ concentration in the platelets and plasma of CKD+AH+DM patients was slightly higher in comparison with other CKD groups. We suppose that several reasons could be behind the higher levels of coenzyme Q_10_ in platelets of CKD+AH+DM patients, such as daily food, drugs, vitamins, and trace elements intake, as well as an adaptation of the antioxidant defense system to chronic oxidative stress. Increased antioxidant CoQ_10_ concentration in diabetic patients may contribute to the self-protection of the body during oxidative stress development [33,34]. These results are supported by other studies in diabetic patients, in children with diabetes, and in rats with experimental diabetes [35,36,37].

The rate of oxygen consumption by intact platelets in the CKD-ALL group positively correlated with CoQ_10-TOTAL_ concentration in platelets (*p* < 0.05, Figure 4). The association between these parameters after the division of the CKD-ALL group into subgroups (CKD+AH and CKD+AH+DM) showed the same trend in both subgroups, but the number of subjects in the subgroups were too small to reach the statistical significance. Conversely, another study with larger group of CKD patients showed a significant positive correlation between CoQ_10-TOTAL_ concentration in platelets and multiple parameters of platelet respiration [38]. These results show the importance to include a higher number of patients in this type of study.

Increased respiration in intact platelets in patients after fluvastatin and atorvastatin treatment, in patients with diabetes mellitus, and in dialyzed patients was found [39]. Opposite results—decreased respiration in intact platelets—were reported in depressive patients [18]. The reduced activity of mitochondrial respiratory chain Complex I was found in the platelets of patients with septic shock and cardiogenic shock and in patients with Parkinson’s disease, while patients with schizophrenia showed an increase in Complex I activity [28,29]. All these platelet mitochondrial results were received by using the high-resolution respirometry method.

Basal mitochondrial oxygen consumption in permeabilized platelets with Complex I-linked substrates (step 1PM) in CKD-ALL patients was slightly decreased vs. the control group (Figure 2). A previous study showed damaged CI-linked respiration in a patient after renal transplantation and strongly impaired CI-linked respiration in a patient with thrombocytopenia [39]. Decreased Complex I-linked respiration in permeabilized platelets in humans after statins treatment has been reported, suggesting the use of the respirometric analysis of mitochondrial function in platelets for studying changes in cellular energy metabolism in patients treated with statins [30]. We suppose that impaired mitochondrial function in patients with nephropathies may be associated with an increased oxidative stress and impaired biosynthesis of coenzyme Q_10_—an essential factor for normal mitochondrial function.

The major sites of mitochondrial ROS generation are at Complex I and Complex III. An impaired mitochondrial respiratory chain and decreased antioxidant defense can result in oxidative stress. In our previous study with a larger number of patients with nephropathy (*n* = 57), we found a significantly increased lipid peroxidation in comparison with the control group, and this negatively correlated with CoQ_10_ concentration in plasma [38]. Significantly higher TBARS values in patients with CKD were found also in other studies [32,40]. In our trial with twenty-seven CKD patients, TBARS were slightly but not significantly increased vs. control data, and the negative correlation with CoQ_10_ concentration in plasma did not reach statistical significance (Figure 5). We are aware that the small number of subjects in the followed groups was the main limitation of our study. Small sample size decreases the statistical power, which makes challenging to detect small changes in evaluated parameters.

## 5. Conclusions

In conclusion, the parameters of platelets mitochondrial respiration showed slight but not significant differences between groups of CKD patients and control subjects. However, the oxygen consumption by intact platelets positively correlated with the concentration of CoQ_10_ in the platelets of CKD patients. We suppose that a reduced CoQ_10_ concentration and increased lipids peroxidation could contribute to decreased platelet mitochondrial function and to the progression of renal dysfunction in patients with CKD diseases. The respiratory rates of platelets obtained by high-resolution respirometry can be used only as a weak biological marker for mitochondrial diagnosis and therapy monitoring in CKD patients. Supplementary studies with more subjects are needed.

## Figures and Tables

**Figure 1 diagnostics-10-00176-f001:**
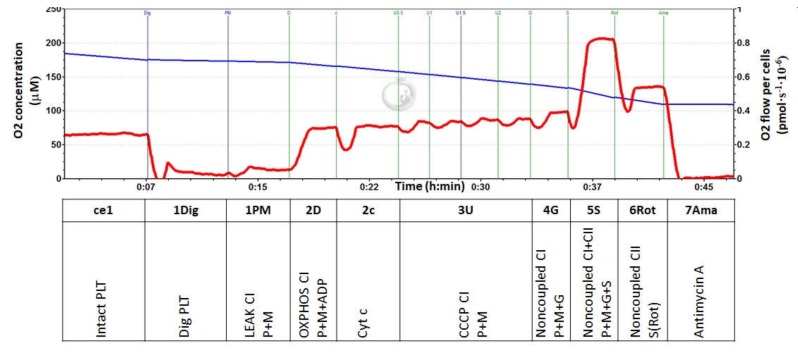
Respirometric analysis of mitochondrial function in human platelets. Legend: The trace from the measurement of platelet (PLT) respiration at 37 °C in a respiration medium MiR05 and 20 mM creatine. Blue line represents oxygen concentration (µM), and the red trace represents oxygen consumption as flow per cells (pmol O_2_ ∙ s^−1^ ∙10^−6^ cells). The modified (substrate-uncoupler-inhibitor-titration) SUIT reference protocol 1 (RP1) [22] includes following steps:

**Figure 2 diagnostics-10-00176-f002:**
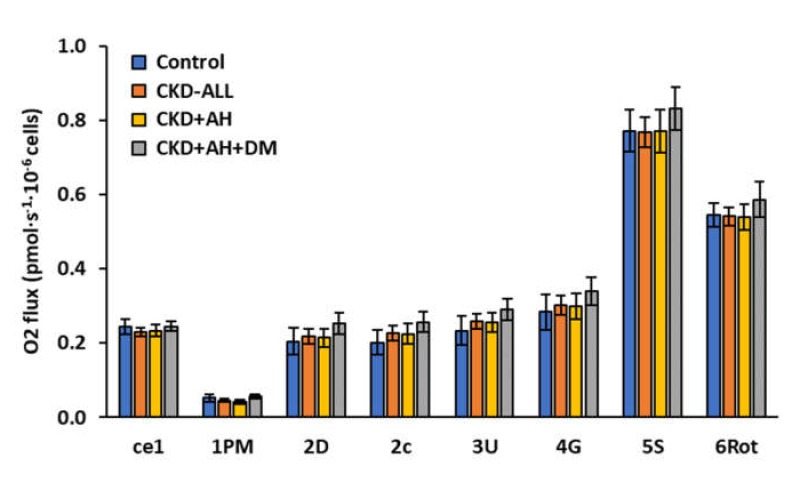
Platelet mitochondrial function in control subjects and groups of CKD patients. Legend: The parameters of mitochondrial respiration in human platelets. The bars show mean ± SEM. The names on the x-axis represent steps in the SUIT RP1—see the legend for Figure 1 in the Methods section. Control—the group of healthy subjects (*n* = 12); CKD-ALL—all patients with chronic kidney disease (*n* = 27); CKD and arterial hypertension (CKD+AH)—the subgroup of CKD-ALL patients with arterial hypertension (*n* = 17); and CKD+AH+diabetes mellitus (DM)—the subgroup of CKD-ALL patients with arterial hypertension and diabetes type 2 (*n* = 10).

**Figure 3 diagnostics-10-00176-f003:**
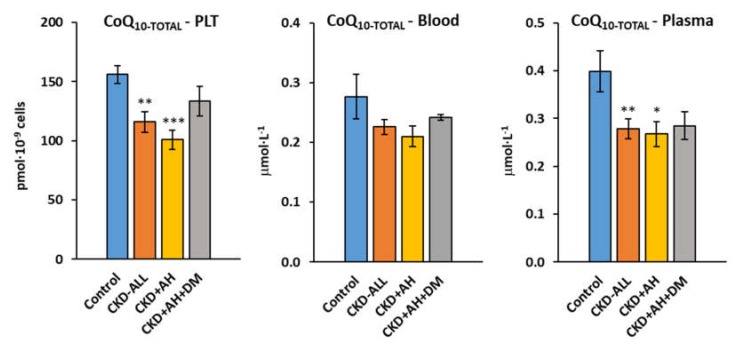
Endogenous coenzyme Q_10-TOTAL_ in platelets, blood, and plasma in control subjects and groups of CKD patients. Legend: coenzyme Q_10_ (CoQ_10_)_-TOTAL_ (ubiquinol and ubiquinone); PLT—platelets; * *p* < 0.05; ** *p* < 0.01; and *** *p* < 0.001. Control—the group of healthy subjects (*n* = 12); CKD-ALL—all patients with chronic kidney disease (*n* = 27); CKD+AH—the subgroup of CKD-ALL patients with arterial hypertension (*n* = 17); CKD+AH+DM—the subgroup of CKD-ALL patients with arterial hypertension and diabetes type 2 (*n* = 10).

**Figure 4 diagnostics-10-00176-f004:**
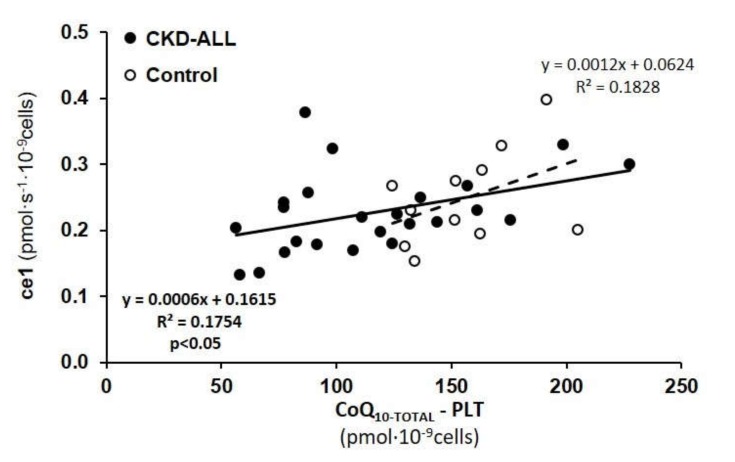
Correlation between CoQ_10-TOTAL_ in platelets and the respiration of intact platelets in control subjects and CKD-ALL patients. Legend: ce1—the rate of oxygen consumption in intact platelet; CoQ_10-TOTAL_—ubiquinol and ubiquinone. CKD-ALL all patients with chronic kidney disease (*n* = 27); Control—the group of healthy subjects (*n* = 12). *p* < 0.05 statistically significant association between CoQ_10-TOTAL_ in platelets and ce1 in CKD-ALL.

**Figure 5 diagnostics-10-00176-f005:**
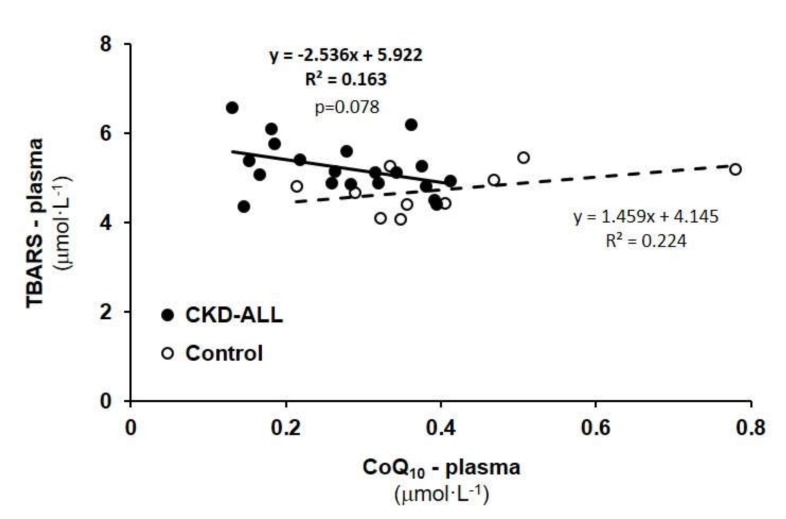
Correlation between thiobarbituric acid reactive substances (TBARS) and plasma CoQ_10-TOTAL_ in control subjects and CKD-ALL patients. Legend: CoQ_10-TOTAL_—ubiquinol and ubiquinone. CKD-ALL—all patients with chronic kidney disease (*n* = 27); Control—the group of healthy subjects (*n* = 12).

**Table 1 diagnostics-10-00176-t001:** Metabolic parameters of human volunteers and groups of chronic kidney disease (CKD) patients.

Groups	CONTROL	CKD-ALL	CKD+AH	CKD+AH+DM
**Table 1a**				
Kidney				
eGFR (ml s^−1^.1.73 m^2^ )	1.328 ± 0.069	0.578 ± 0.043 ***	0.574±0.053 ***	0.583±0.074 ***
Creatinine (µmol L^−1^)	70.7 ± 5.1	194.3 ± 17.0 ***	198.5 ± 19.6 ***	187.3 ± 31.3 **
Uric acid (µmol L^−1^)	287.4 ± 13.7	370.8 ± 25.5 *	346.8 ± 25.9	409.2 ± 47.9 *
**Table 1b**				
Blood				
Hgb (g L^−1^)	140.5 ± 2.2	129.9 ± 4.4	132.9 ± 4.8	124.7 ± 8.3
CRP (mg L^−1^)	1.73 ± 0.28	7.23± 1.26 ***	6.80 ± 1.44 **	7.83 ± 2.30 *
Glucose (mmol L^−1^)	5.78 ± 0.17	6.19 ± 0.35	5.79 ± 0.24	6.87 ± 0.81
**Table 1c**				
Lipids				
TAG (mmol L^−1^)	1.16 ± 0.12	1.78 ± 0.18	1.71 ± 0.21	1.92 ± 0.35
LDL-Chol (mmol L^−1^)	3.75 ± 0.13	3.55 ± 0.21	3.50 ± 0.26	3.65 ± 0.34
HDL-Chol (mmol L^−1^)	1.50 ± 0.07	1.31 ± 0.07	1.33 ± 0.09	1.27 ± 0.10
Chol-Total (mmol L^−1^)	5.70 ± 0.15	5.5 ± 0.31	5.30 ± 0.35	6.00 ± 0.56
**Table 1d**				
Liver				
AST (kat L^−1^)	0.371 ± 0.028	0.405 ± 0.037	0.393 ± 0.037	0.421 ± 0.070
ALP (kat L^−1^)	0.297 ± 0.017	0.863 ± 0.148 ***	0.930 ± 0.203 *	0.756 ± 0.204 *
GMT (kat L^−1^)	0.751 ± 0.130	0.530 ± 0.066	0.505 ± 0.082	0.566 ± 0.111

Data of all groups are presented as mean ± SEM and statistically evaluated in comparison with control group: * *p* < 0.05; ** *p* < 0.01; and *** *p* < 0.001.

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
