# Peer review of "Platelet Mitochondrial Respiration, Endogenous Coenzyme Q_10_ and Oxidative Stress in Patients with Chronic Kidney Disease"

_diagnostics, 2020, doi:10.3390/diagnostics10030176_

Round 1
Reviewer 1 Report
The manuscript by Anna et al, “Platelet Mitochondrial Respiration, Endogenous Coenzyme Q10 and Oxidative Stress in Patients with Chronic Kidney Disease” is written well with the hypothesis to determine the mitochondrial bioenergetic functions in cells using a modified method of an established HRR method. The authors also used other common UV method to detect CoQ10 levels in platelet, blood and plasma, where they actually see some significant changes in the CoQ10 levels in platelets and plasma. Though the manuscript has some good information I have several concerns as below,
- It is known the lymphocytes and monocytes play a major role in CKD progression, why the author chooses to see mitochondrial functions only on platelets
- Methods, line 267, The author has divided the CKD group in 3, CKD-ALL, CKD+AH, CKD+DM, eventually all subjects are treated with relevant conventional therapy. However, all the data presented are post treatment, where I don’t see a major difference in most of the assays. Without a pre-treatment data of a patients, it is hard to get a reliable information from post-treated CKD patients for establishing a new improved HRR method.
- Table 1b, line 78, how does the patient’s glucose level in CKD+DM has almost a normal range of glucose levels, what type of DM they have mostly?
- Figure 2, line 130, why does the base CoQ10 levels in whole blood is lower than that plasma itself? So, what contributes the higher level of CoQ10 in plasma, but not in the whole blood? Did the author measure the CoQ10 levels in WBCs?
- Line 138, how does CKD-ALL have a positive correlation with O2 consumption by platelets, where the author doesn’t see when they separate the subjects based on the AH and DM? How does the data differ when subjects are separated?
- Discussion sections are long with repeating results part, the subtitles may be removed and discussion should be shortened to make more clear direct points.
Author Response
Gvozdjáková – Answer to reviewer 1:
1./ Introduction was improved, with a new reference included
2./ Aim of the trial was reworded
3./ The methods are described in more detail
4./ Results are clearly presented
5./ Conclusions are modified
6./ We aimed to study mitochondrial function in CKD on platelets as surrogates for tissue biopsy. Immune cells are sensitive to pro-inflammatory conditions, therefore we choosed platelets and not peripheral blood mononuclear cells (lymphocytes+monocytes) for our study. Most of published references used platelets for mitochondrial study (see references: 10, 14, 16, 18, 19, 20, 21).
7./ Methods: all CKD patients were with chronic kideny diseases treated by conventional therapy. Blood samples were collected in the morning, before taking the conventional therapy.
8./ Table 1b: In CKD+AH+DM patients - glucose level was the highest in comparison with other groups. The patients were treated with conventional antidiabetic therapy. Group CKD+DM was renamed to CKD+AH+DM. All of these patients had arterial hypertension and diabetes type 2.
9./ Figure 2: CoQ10 is present in all cellular membranes as well as in high- and low-density lipoproteins present in plasma. Red blood cells have relatively low content of CoQ10. The concentration of CoQ10 in blood depends on CoQ10 in plasma, CoQ10 in blood cells and the ratio of blood cells volume to plasma volume. The concentration of CoQ10 in plasma is probably higher than in blood cells – therefore the concentration of CoQ10 in plasma is higher than in blood. We did not measure CoQ10 levels in WBC.
10./ The rate of oxygen consumption in intact platelet positive correlated with the concentration of CoQ10 in platelets in CKD-ALL group (N=27). The division into subgroups showed the same trend, but the number of subjects in the subgroups was too small to reach the statistical significance.
We are aware that the low number of subjects in all followed groups including the control group was the limiting factor in this study. Unfortunately, due to long-lasting technical problems, we cannot currently add more subjects to this study.
11./ Discussion is shorter, reworded, without sub-titles and repeated results.
Reviewer 2 Report
The manuscript entitled “Platelet Mitochondrial Respiration, Endogenous Coenzyme Q10 and Oxidative Stress in Patients with Chronic Kidney Disease” of Gvozdjakova et al. suggests usage of high-resolution respirometry method to test parameters of mitochondrial respiration of platelets of patients with Chronic Kidney Disease for diagnosis. The idea is very exciting as this method could be only slightly invasive.
The major concern about the manuscript is that this study did not show any significant changes in the respiration parameters of platelets mitochondria. At this point, I do not see that this method can be used in any kind of diagnostic. Moreover, the authors claim that CoQ10 can contribute to slightly decrease parameters of respiration in platelets. This conclusion made from a simple correlation with not significantly decreased oxygen consumption. The quality of experiments is acceptable but these experiments do not support the conclusion.
Minor concerns:
Figure legends should be expanded. All the abbreviations used in the picture should be explained.
In figure 4 one control point seems to be out of the range of data set. Correlation without this point should be analyzed or more control data should be added.
Author Response
Gvozdjáková – Answer to reviewer 2:
- Aim of the trial was reworded
- The methods are described in more detail
- Conclusions are modified
- Figure: all abbreviations in the pictures are explained
- Figure 4: there was no significant correlation between TBARS and plasma CoQ10-TOTAL in control subjects with or without the one control point.
We are aware that the low number of subjects in all followed groups including the control group was the limiting factor in this study. Unfortunately, due to long-lasting technical problems, we cannot currently add more subjects to this study.
Round 2
Reviewer 1 Report
The authors addressed all concerns raised by the reviewers and the revised manuscript is significantly improved.
Reviewer 2 Report
The authors addressed all the concerns reflected in my previous report. This study definitely contributes to diagnostic development but needs more subject as it claimed by authors.